# Copy number variation of two begomovirus acquired and inoculated by different cryptic species of whitefly, *Bemisia tabaci* in Okra

Mritunjoy Barman[1☯], Snigdha Samanta[1☯], Swati Chakraborty[2], Arunava Samanta[1], Jayanta Tarafdar[2,3]*

**1** Department of Agricultural Entomology, B.C.K.V, Mohanpur, West Bengal, India, **2** Department of Plant Pathology, B.C.K.V, Nadia, West Bengal, India, **3** Directorate of Research, B.C.K.V, Kalyani, India

☯ These authors contributed equally to this work.
* tarafdar.jayanta@bckv.edu.in

**Data Availability Statement:** The sequences generated in this study are available in the NCBI GenBank (accession numbers: MZ973007,

## Abstract

The whitefly, *B.tabaci* is a major pest of agricultural crops which transmits begomovirus in a species-specific manner. Yellow vein mosaic disease (YVMD) and okra leaf curl disease (OLCD) caused by distinct begomovirus are a major limitation to production of okra in India. In this framework the present investigation reports, for the first time, comparative study of begomovirus species viz. yellow vein mosaic virus (YVMV) and okra enation leaf curl virus (OELCuV) ingested and egested by two cryptic species (Asia I and Asia II 5) of *B.tabaci* at different time interval using detached leaf assay. A gradual increase of both virus copies were observed with increased feeding exposure in Asia I and Asia II 5. Both the genetic groups of whitefly could acquire the viruses within just 5 minutes of active feeding however, a significant amount of variation was noted in virus uptake by the both. At 24 hours of active feeding Asia II 5 acquired more of YVMV whereas, Asia I ingested more OELCuV. Similarly, the genetic group acquiring higher titre of virus egested higher amount during inoculation period. On the whole, it can be presumed that Asia I is a more effective transmitter of OELCuV whereas, Asia II 5 of YVMV further suggesting increased risk of virus pandemics (both YVMV and OELCuV) in regions where Asia I and Asia II 5 is dominant.

## Introduction

The whitefly, *Bemisia tabaci*, an economically important agricultural pest that causes huge damage to crops worldwide both directly and as a vector of nearly 120 geminivirus, which includes Begomovirus, Crinivirus, Closterovirus etc. [1]. *B.tabaci* is listed among the top 100 dreadful alien invasive species [2]. The global distribution of this species ranges from tropical, subtropical, and temperate regions [3]. *B.tabaci* is considered to be a highly variable species complex having their own genetic characteristics which differed in virus transmission ability, host plant preferences, fecundity and even insecticide resistance [3]. The interactions between plants, whitefly and begomoviruses have been drawing attention of researches for the last five decades [4]. Based on genome organization, host diversity and vector specificity, the members

MZ973008, MZ772932, MZ772927, OL743532, OL743533).

**Funding:** The authors received no specific funding for this work.

**Competing interests:** The authors have declared that no competing interests exist.

in the family Geminiviridae were earlier classified into four genera: Mastrevirus, Curtovirus, Topocuvirus and Begomovirus [5]. According to the recent report of International Committee on Taxonomy of Viruses (ICTV), the Geminiviridae family constitutes of nine genera embracing >360 species. Moreover, recent metagenomic studies have led to the establishment of five new genera under the family Geminiviridae (Citlodavirus, Maldovirus, Mulcrilevirus, Opunvirus, and Topilevirus [6]. Viruses in the genera Becurtovirus, Capulavirus, Curtovirus, Eragrovirus, Grablovirus, Mastrevirus, Topocuvirus and Turncurtovirus have monopartite genomes, whereas those in the genus Begomovirus have mono- or bipartite genomes [7].

Begomovirus, transmitted exclusively by whiteflies in a persistent- circulative manner has emerged as a serious threat to the production of many vegetable crops [8, 9]. Plants infected with begomovirus exhibit typical symptoms like leaf curling, vein yellowing, yellow mosaic, stunting and vein thickening. The whitefly ingests these begomovirus by injecting its stylet in the vascular tissue [10]. Post ingestion the virus particles are translocated through the digestive system of the vector into the haemolymph. Subsequently the virus gets stored in the salivary glands from where it is egested into the phloem. Transmission of begomovirus like *tomato yellow leaf curl virus* (TYLCV), *chilli leaf curl virus* (ChiLCV) have been well documented in crops like tomato, chilli, pumpkin etc. However, not much emphasis is laid on *yellow vein mosaic virus* (YVMV) and *okra enation leaf curl virus* (OELCuV) both monopartite virus and a major threat to okra cultivation in India [11–14]. So far ten cryptic species of the *B.tabaci* complex have been recorded in India. The *B.tabaci* fauna distributed across the Indian subcontinent with Asia I and Asia II 5 predominantly occurring in eastern provinces [15]. Studies reveal that the prevalence of different genetic groups of *B.tabaci* adds to the complexity as transmission efficacy of the virus is different for different genetic group/ strain [16].

While there are researches highlighting the epidemiology and transmission of these begomovirus by *B.tabaci* species complex, the number of copies ingested or egested by individual whitefly has not been studied yet. In this light the present study was designed to measure the viral copies (YVMV and OELCuV) acquired and inoculated by Asia I and Asia II 5 whitefly using leaf detach assay. The findings will have significant implications in better understanding of the virus epidemiology while providing a comparative idea regarding the virus-vector relationship of two genetic groups of whitefly.

## Materials and methods

### Insect vector

Cultures of two different cryptic species (Asia I and Asia II 5) of *B. tabaci* complex were initially collected from two localities of Bengal province and maintained on eggplant (var. Samrat) for establishing a uniform population. For identification, the homogenous population was characterized by using mitochondrial cytochrome oxidase subunit I (mt-COI). The population was kept in regulated environmental conditions at $26 \pm 2°$ C, $70 \pm 10\%$ RH, and 16 hrs light/8 hrs dark photoperiod in Molecular biology laboratory, Directorate of Research, BCKV and considered as stock culture. The purity of each culture was carefully maintained and further confirmed by sequencing of 5–10 whitefly individuals of both genetic group using mt-COI gene at every 15 days interval. During the course of experiment the whitefly populations were repetitively tested in PCR for confirmation of its aviruliferous status. The newly emerged adult females were collected by using an aspirator for further experiments.

### Virus isolates

The pure culture of both YVMV and OELCuV isolates has been maintained separately at molecular biology laboratory, BCKV. Both the begomovirus were maintained in separate okra

plants (var. MONA 002) by *B. tabaci*-inoculation in two different insect proof cages. Begomoviruses maintained in separate cages were confirmed by using YVMV and OELCuV gene specific primer. Cross check of the inoculated plants were done during the course of experimentation to avoid any mixed infection.

## Identification of *B. tabaci* and begomoviruses

Extraction of DNA from the samples (both whitefly and inoculated okra plant) was conducted with the help of Genomic DNA Isolation Kit (Sigma-Aldrich, St Louis, USA). DNA samples were checked for its purity and concentration using Nano Drop 1000 spectrophotometer (Thermo Fisher Scientific, Wilmington, DE, USA) and the eluted product was stored at −80˚- for further use. Molecular identification of both homogeneous whitefly population and begomoviruses were done by using specific primer (mt-COI gene for whitefly and coat protein for Begomoviruses) listed in Table 1.

Each PCR reaction contained 2μL DNA(~40 ng for whitefly and ~100 ng for Okra plants), Taq DNA polymerase (3U/μL) (Bioline, USA), 2.5μL Taq buffer (10X) (Bioline, USA), 1μL dNTPs (2.5mM) (New England Biolab), 3μL MgCl2(15mM) (Bioline, USA), 1μL Forward Primer (10mM), 1μL Reverse Primer(10mM) and sterile water to make up total volume of 25μL. The reaction involved denaturation at 94˚C for 30s and annealing at different temperatures (54˚C for whitefly characterization, 59˚C for YVMV and OELCuV) for a time period of 30 s with 35 number of cycles. Extension was carried out at 72˚C for 40 s with the final extension for 5 min at 72˚C. Each PCR products were resolved on 1% agarose gel and visualized in a gel documentation system. PCR products were further purified using gel elution and purification kit (HiPurA^TM PCR Product and Gel Purification Combo Kit). Purified PCR products were cloned with the help of pGEM T-easy Vector and transformed into DH5α E. coli cells (Promega, Madison, USA) [17]. Plasmid DNA was isolated using Wizard Plus SV Minipreps DNA Purification System (Promega) and then sent for Sanger dideoxy sequencing. Furthermore, the sequences were processed by using BioEdit and BLASTn followed by submission to the NCBI database. To identify the different genetic groups of whitefly a phylogenetic tree was built with the help of maximum likelihood method (Kimura 2-parameter model) having 1000 bootstraps replications [18].

## Acquisition and inoculation of begomoviruses by *B. tabaci*

Individual adult female of *B. tabaci* with a maximum age of 24 hrs was used for virus acquisition and inoculation. Insect breeding dish (65 mm d, 10 mm h) was used and 5 ml of 1% agar-agar was poured in the bottom half. Apical symptomatic leaves of YVMV and OELCuV (50-day old plants) were collected and petioles were inserted in the solidified agar-agar for

**Table 1. List of primer pairs used in the study.**

| Primer name | Primer sequence (5'-3') | Melting temperature (˚C) | Primer name | Primer sequence (5'-3') | Amplicon size (bp) | Target region |
|---|---|---|---|---|---|---|
| C1-J- 2195 | TTGATTTTTTGGTCATCCAGAAGT | 54˚C | TL2-N -3014 | CCAATGCACTAATCTGCCATATTA | 840 bp | mt-COI |
| BYVMV (CP) F | TTTCGGATGTTACGCGTGGA | 59˚C | BYVMV (CP) R | TGCTAGCATGGGTACAAGCC | 800bp | BYVMV Coat protein |
| OELCuV (CP) F | ATCGTCATTTCTACCCCCGC | 59˚C | OELCuV (CP) R | CCTCCTGTAACTGTCGCCTG | 800bp | OELCuV Coat protein |
| BYVMV(CP) RT-F | GCAACTTTTGTCGCAGGATT | 59˚C | BYVMV(CP) RT-R | ATAGGCCTGTTTGTCCATGC | 120bp | BYVMV Coat protein |
| OELCuV (CP) RT-F | GCACCCCCTACGATTTCCAG | 56˚C | OELCuV (CP) RT-R | ACAAGCATACTGTCCTCCTG | 80 bp | OELCuV Coat protein |

virus acquisition by individual *B. tabaci*. Simultaneously, the virus copy number present in source plants itself were also estimated. For inoculation, leaf of 50-day old virus-free okra plant (var. MONA 002) was used for virus inoculation by both the genetic group of *B. tabaci*. To maintain homogeneity leaves of same size and age (approximately 10 square cm) were considered throughout the experiment. A single aviruliferous whitefly, of each cryptic species (Asia I and Asia II 5) were released parallelly on the virus infected leaves for acquisition of the virus. For virus inoculation, individual *B. tabaci* of different groups was released separately on YVMV and OELCuV infected leaves for 24 hrs to virus acquisition and further shifted to virus-free leaves inside the insect breeding dishes. Constant monitoring within the dishes was done to ascertain whether *B. tabaci* started feeding on leaves. Both the acquisition and inoculation sets were kept in dark (26 ±2°C, and 70 ±10% relative humidity) and *B. tabaci* adults were collected from both setups at an interval of 1 min, 5 min, 15 min, 1 hr, 2 hrs, 6 hrs, 12 hrs, and 24 hrs after release. The experiment setup were maintained separately for YVMV and OEL-CuV isolates with three replications (S1 Fig).

## Standardization of real-time PCR

Real-time PCR primers for YVMV and OELCuV were designed based on the coat protein (CP) sequences of the viruses (Table 1). Prior to final set up in real-time PCR a conventional PCR was performed to validate the newly designed primers. The PCR program was carried out in a total volume of 25 μl, containing 2 μl of template DNA (~40 ng for *B. tabaci* and ~100 ng for plants), 12.5 μl of PCR master mix, 8.5 μl of molecular-grade water, and 1 μl each of forward and reverse primer. The PCR was performed in Veriti 96-well Thermal Cycler (Applied Biosystems) with one cycle of initial denaturation at 95°C for 3 min, 40 cycles of denaturation at 95°C for 40 s, annealing at 56°C for OELCuV and 59°C for YVMV for 30 sec. Extension was carried out at 72°C for 40 s followed by a final extension at 72°C for 10 min. The PCR products were resolved on 1% agarose gel and visualized in a gel documentation system. Real-time PCR was performed in using the Agilent Technologies Stratagene Mx3000P Sequence Detection System with 20 μl reaction volume consisted of a 10 μl 2x SYBR Green qPCR Master Mix (Thermo Scientific), 2 μl template DNA (~100 ng for plant and ~40 ng for *B. tabaci*), and 1 μl (10 pmol) of each forward and reverse primer. The CT values (from three biological replicates) obtained were used for calculating the mean CT and standard error of mean (SEM) with the aid of Microsoft Excel software.

## Preparation of standard curve

Total DNA was isolated from the test plant samples using plant DNA extraction kit (Thermo Fisher Scientific). The quantity and quality of the DNA was determined by Nano Drop 1000 spectrophotometer (Thermo Fisher Scientific, Wilmington, DE, USA). For absolute quantification of the virus (YVMV and OELCuV), a standard curve was generated using partial coat protein gene of about 120bp and 80bp size respectively. The PCR amplified product of YVMV and OELCuV were cloned with pJET1.2 vector using JET PCR Cloning Kit (Thermo Scientific) and transformed into DH5α E. coli cells. A ten-fold serial dilution of the linearized plasmid DNA was used for real-time PCR. The real-time PCR was performed in 20 μl reaction volume considering serially diluted plasmid DNA as templates. The serial dilution of the plasmid DNA were used for constructing two different standard curve each of YVMV and OELCuV.

## Quantification of begomovirus copies

For quantification, *B. tabaci* cryptic species (exposed to YVMV and OELCuV) and inoculated leaves were collected at different exposure period and DNA was extracted as described above.

The mean CT values were fitted into their corresponding standard curves for absolute quantification of YVMV and OELCuV using the formula given by Krieg [19].

## Data analysis

Absolute quantification of YVMV and OELCuV in test samples (*B. tabaci* cryptic species and inoculated leaves) at different periods of feeding was calculated by operating the default setup of MxPro qPCR software on Agilent Technologies Stratagene Mx3000P Sequence Detection System. The log of virus copy number ingested and egested by individual *B. tabaci* cryptic species was modelled using logarithmic function separately in Excel.

## Results

### Characterization of cryptic species of *B. tabaci*

Several DNA-based techniques have been exploited for proper identification of *B. tabaci* cryptic species [20, 21]. Nonetheless, sequence analysis of mitochondrial cytochrome oxidase I (mt-COI) gene have been most widely accepted [22–25]. In the current study running culture of two different homogenous population of *B.tabaci* were identified by using the primer pair (C1-J-2195 F/ L2-N-3014 R) of the universal mt-COI gene. Based on the previously known sequences in the GenBank database, a phylogenetic tree was constructed by using maximum likelihood phylogram (Fig 1). The phylogenetic analysis of the determined COI sequences assured that the populations belonged to two different cryptic species Asia I and Asia II 5. The sequence can be retrieved using the GenBank Accession No. MZ973007, MZ973008 for Asia I and MZ772932, MZ772927 for Asia II 5.

### Characterization of the begomoviruses

PCR with YVMV and OELCuV -inoculated samples produced ~800 bp products. The sequencing results of the products could generate 770 nt (Accession No. OL743532) and 780 nt (Accession No. OL743533) sequences for YVMV and OELCuV, respectively. From BLASTn analysis, we obtained 100% similarity with other YVMV and OELCuV coat protein sequences available in NCBI.

### Standard curve

A relative quantification method could not be used because of the absence of any control group; hence, absolute quantification was considered to be the best way to compare the viral load against different treatments. To estimate the Begomovirus (YVMV and OELCuV) copy number acquired and egested by individual whitefly, Ct values were generalized to a calibration curve obtained from amplifying serial dilutions of plasmid standards using qPCR. The resultant Ct values were plotted against the known copy numbers of the standard sample. The standard curve covered a linear range of five different orders of magnitude. Primer pair, BYVMV (CP) RT-F and BYVMV (CP) RT-R; OELCuV (CP) RT-F and OELCuV (CP) RT-R produced prominent bands of 120 and 80 bp for YVMV and OELCuV, respectively. Both the primer pairs did not produce any secondary peak in the melting curve analysis in real-time PCR (Figs 2 & 3). The specific melting temperature for YVMV and OELCuV products were around 78˚C and 84˚C respectively. The standard curve for both the viruses showed an ideal high amplification efficiency of 110.4% and 105.3% respectively representing the ideal conditions for absolute quantification. The correlation coefficient ($R^2$) of both the standard curve was noted to 0.998, signifying that this assay could be used to quantify target DNA virus (both acquired and egested) by individual whitefly.

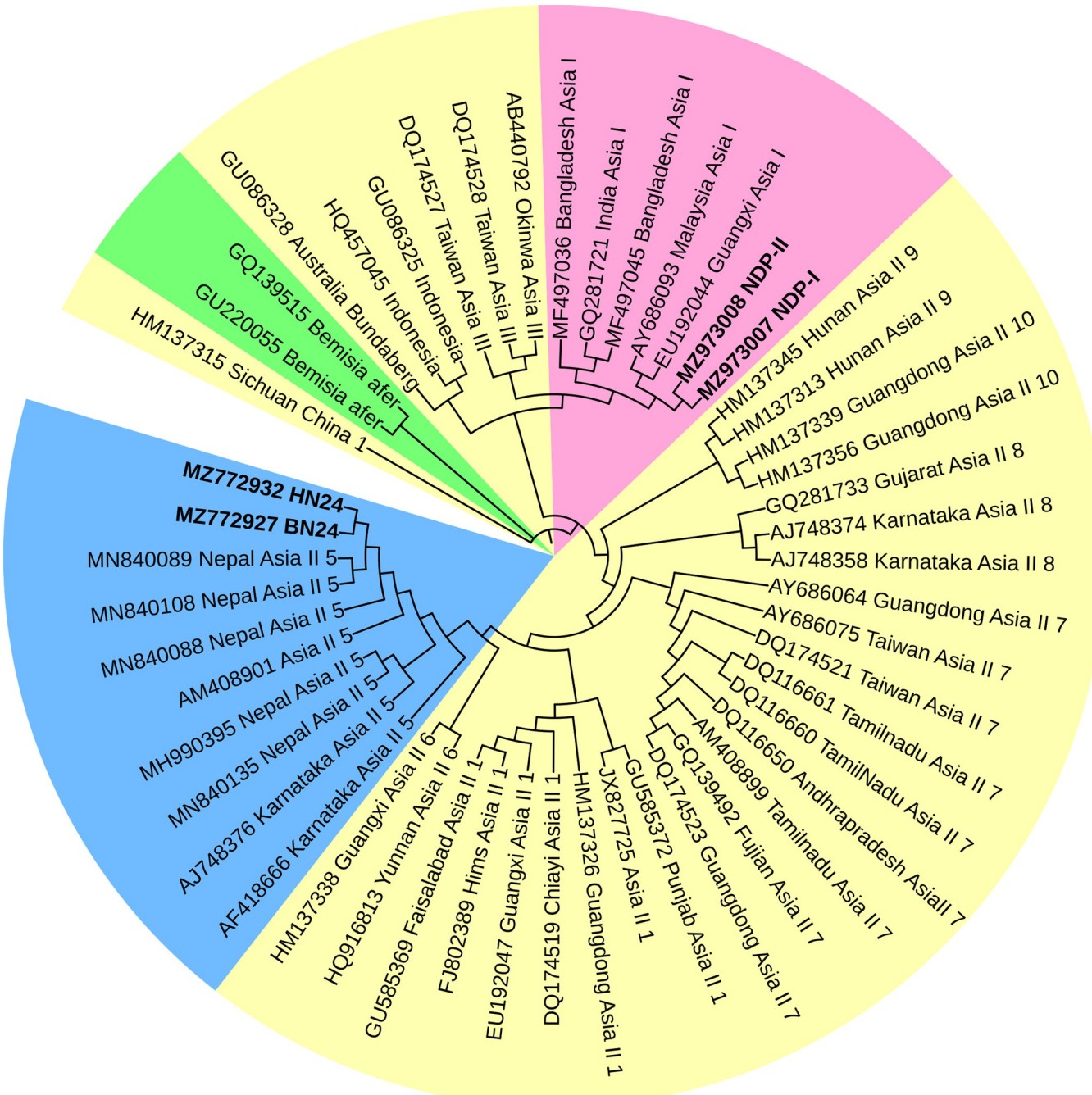

**Fig 1. Phylogenetic tree of *B. tabaci* cryptic species identified based on cytochrome oxidase subunit I (COI) sequences.** The samples from the study are indicated by bold text in the tree; all other sequences were obtained from the GenBank database. Pink and blue coloured shade represent sequences of Asia I and Asia II 5 respectively, whereas, yellow shade represent the other cryptic species of *B.tabaci*. *Bemisia afer* sequences (green shade) was taken as an out-group.

## Comparison of begomovirus titre acquired by Asia I and Asia II 5

The absolute quantification (qPCR) method was used to detect copies of the YVMV and OEL-CuV acquired by individual whitefly sample (Asia I and Asia II 5 genetic group) from the source leaves at different feeding periods. Subsequently, both the cryptic species of *B.tabaci* were classified into positive (viruliferous) or negative (non-viruliferous) according to the

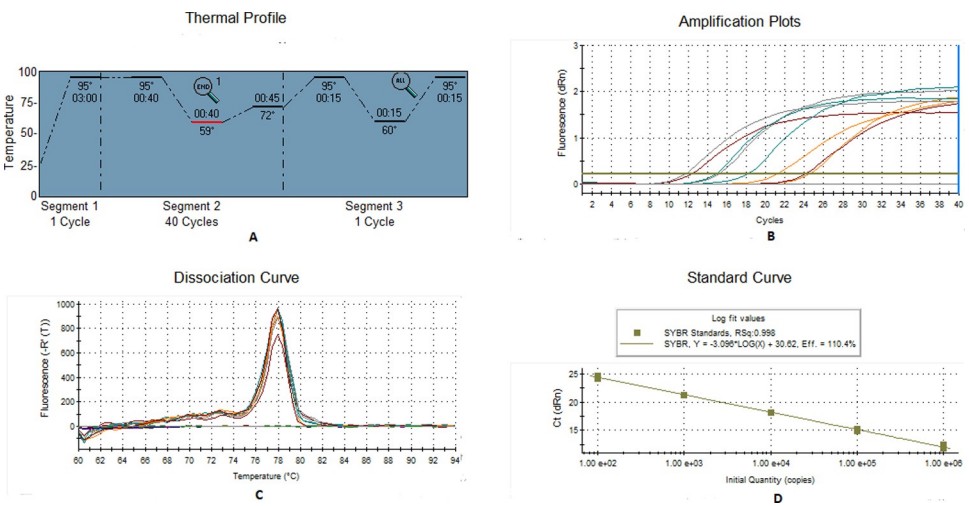

**Fig 2.** Real-time PCR analysis of YVMV indicate the specificity of the reactions (A) Thermal profile (B) PCR amplification plots (C) dissociation curve and (D) standard curves of serially diluted linearized plasmids obtained using SYBR® Green chemistry. The specific melting temperature for YVMV products was around 78°C. Standard curves show a linear relationship between initial viral copy number per microliter on X-axis and Ct values on Y-axis. Each concentration was replicated twice. The equation of the straight line and the coefficient of correlation ($R^2$) are mentioned on the graph.

detectable targeted gene (coat protein for YVMV and OELCuV) levels. The viral copy numbers present in both the cryptic species were represented as $\log_{10}$ converted values. The virus copies in the source leaves used for acquisition of YVMV and OELCuV by *B. tabaci* was measured to be 5.98 and 5.93 copies per microliter respectively. A total of 8 feeding periods starting from 1 min upto 24 hr were tested for both the cryptic species of *B.tabaci*. No viral load could be detected after 1 min of active feeding and considered as qPCR negative, whereas the

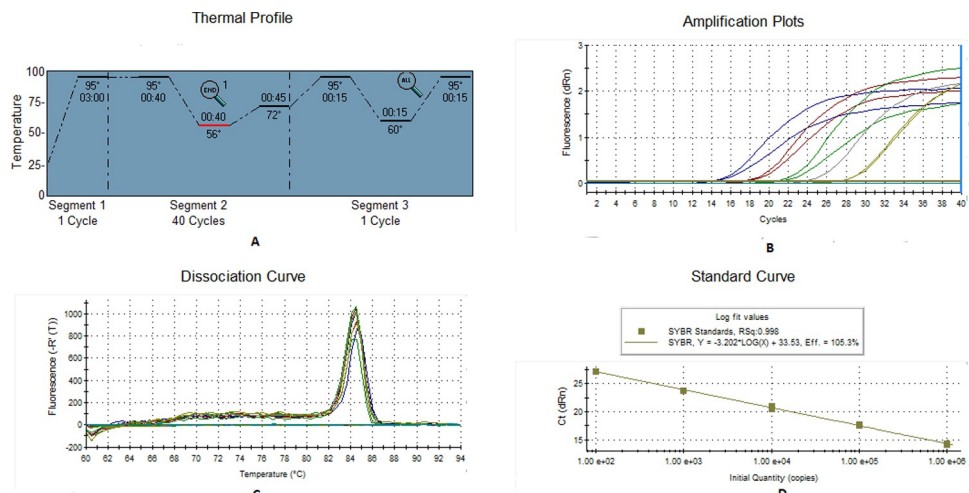

**Fig 3.** Real-time PCR analysis of OELCuV indicate the specificity of the reactions (A) Thermal profile (B) PCR amplification plots (C) dissociation curve and (D) standard curves of serially diluted linearized plasmids obtained using SYBR® Green chemistry. The specific melting temperature for YVMV products was around 84°C. Standard curves show a linear relationship between initial viral copy number per microliter on X-axis and Ct values on Y-axis. Each concentration was replicated twice. The equation of the straight line and the coefficient of correlation ($R^2$) are mentioned on the graph.

**Table 2. Copies of YVMV and OELCV ingested by individual *B. tabaci* (Asia I and Asia II 5) genetic group at different feeding exposure.**

| Feeding exposure | Mean (log$_{10}$ YVMV copies µl$^{-1}$) | | Mean (log$_{10}$ OELCV copies µl$^{-1}$) | |
|---|---|---|---|---|
| | Asia I | Asia II 5 | Asia I | Asia II 5 |
| **1 min** | / | / | / | / |
| **5 min** | 2.05±0.08,a | 2.13±0.02,a | 2.32±0.10,a | 2.19±0.25,a |
| **15 min** | 2.25±0.09,a | 2.19±0.21,a | 2.77±0.06,b | 2.37±0.05,a |
| **1 hr** | 2.44±0.10,b | 2.30±0.06,b | 3.06±0.15,b | 2.80±0.13,b |
| **2 hrs** | 2.78±0.10,b | 3.19±0.15,c | 3.74±0.15,c | 2.98±0.06,b |
| **6 hrs** | 3.97±0.05,c | 3.85±0.18,d | 4.04±0.08,d | 3.51±0.09,c |
| **12hrs** | 4.00±0.10,c | 4.38±0.05,e | 4.15±0.01,d | 3.78±0.15,c |
| **24hrs** | 4.49±0.16,d | 5.12±0.09,f | 5.14±0.11,e | 4.74±0.07,d |

remaining 7 time points were qPCR-positive. After 1 hr of active feeding titre level of both YVMV and OELCuV were estimated to be 2.44 and 3.06 copies per microliter respectively in case of Asia I group, whereas for the same amount of time, Asia II 5 contained 2.30 and 2.80 copies per microliter, of both the viruses respectively (Table 2). From this result it is evident that Asia I genetic group of *B.tabaci* acquired more titre of both the viruses when compared to that of Asia II 5 during the initial feeding period. A steep increase of both virus copies were observed with increased feeding exposure in the two genetic groups of whitefly. Amount of YVMV titre in Asia I group was estimated to be 4.00 copies per microliter at 12 hrs followed by a peak of 4.49 copies per microliter after 24 hrs of feeding (Fig 4A and 4B). On the contrary, in Asia II 5, maximum titre of YVMV was noted after 24 hrs of feeding, with 5.12 copies per microliter (1.14-fold times higher as compared to Asia I genetic group). In the case of OEL-CuV, the virus copies accumulating in Asia I group was always higher than Asia II 5 throughout the feeding period. Maximum titre of OELCuV was noted after 24 hrs of continuous feeding, with 5.14 copies per microliter, which was 1.05-times higher as compared to Asia II 5 genetic group (4.74 copies per microliter after 24 hrs.).

## Comparison of begomovirus titre egested by Asia I and Asia II 5

Inoculation of Begomovirus by each of the whitefly was quantified in detached okra leaves. No begomoviruses titre was quantified by both the genetic group after 1 min of active feeding and were consider as qPCR negative. However, both the viruses could be detected in real-time PCR after 5 min in both the cryptic species egested okra leaves. After 5 min of continuous feeding YVMV copy numbers egested by individual Asia I and Asia II 5 groups were 2.08 and 2.11 copies per microliter, respectively in detached okra leaves (Table 3). The virus copy number increased in the leaves with increased feeding exposure. The virus copies increased to 2.28 and 2.24 copies per microliter, respectively for both Asia I and Asia II 5 at 1hr post feeding (Fig 4C). So, it is clear that after initial phases of egestion Asia I group of *B.tabaci* inoculated more titre of YVMV as compared to Asia II 5. However, the difference in titre was non-significant. Alternatively, in case of Asia II 5 maximum YVMV titre in detached okra leaves was noted 24 hrs post feeding, with 4.71 copies per microliter, around 1.21-fold higher as compared with Asia I (3.89 copies per microliter) genetic group.

Similarly, after 5 min of constant feeding OELCuV copy numbers egested by individual Asia I and Asia II 5 groups in detached okra leaves were 2.14 and 2.01 copies per microliter, respectively (Table 3). The virus copies increased to 2.51 and 2.18 copies per microliter, for both Asia I and Asia II 5 respectively at 1hr post feeding. Hence, there are similarities with that of YVMV i.e. after initial phases of egestion Asia I group of *B.tabaci* inoculate more titre of

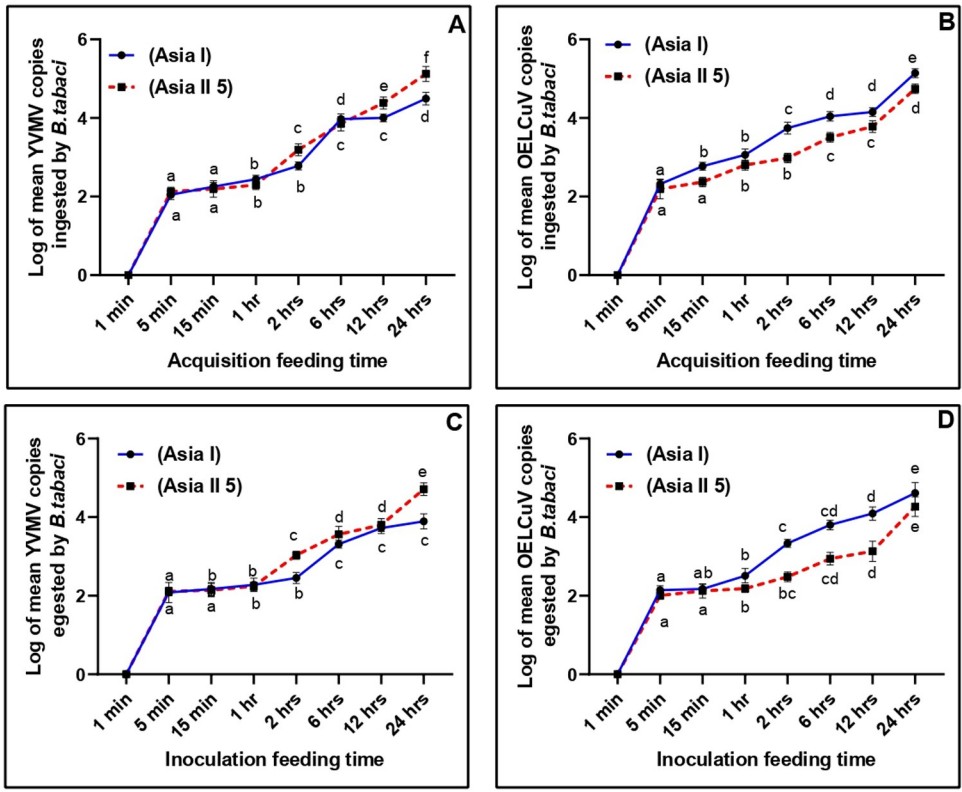

**Fig 4.** Dynamics of YVMV and OELCuV copies ingested (A and B) and egested (C and D) by individual Asia I and Asia II 5 genetic group of *B. tabaci*. Acquisition/ Inoculation feeding time is expressed on X-axis and log of virus copy number is plotted on Y-axis. YVMV and OELCuV copies were estimated at 1 min, 5 min, 15 min, 1 hr, 2 hrs, 6 hrs, 12 hrs, and 24 hrs of feeding. The different letters indicate statistically significant differences between the treatments. The bars represent the standard error of mean (± SEM).

OELCuV than Asia II 5. The viral load reached its highest level after 24 hrs of feeding in case of Asia I genetic group, with 4.61 copies per microliter, i.e. 1.07-fold higher as compared with Asia II 5 (4.27 copies per microliter) genetic group (Fig 4D).

## Discussion

*B.tabaci* is one of most dreadful pests which serves as a vector to large no. of plant viruses in the genera *Begomovirus*, *Carlavirus*, *Ipomovirus* and *Closterovirus* [8]. Amongst them

**Table 3. Copies of YVMV and OELCV egested by individual *B. tabaci* (Asia I and Asia II 5) genetic group at different feeding exposure.**

| Feeding exposure | Mean (log$_{10}$ YVMV copies μl$^{-1}$) | | Mean (log$_{10}$ OELCV copies μl$^{-1}$) | |
|---|---|---|---|---|
| | Asia I | Asia II 5 | Asia I | Asia II 5 |
| 1 min | / | / | / | / |
| 5 min | 2.08±0.26,a | 2.11±0.07,a | 2.14±0.11,a | 2.01±0.10,a |
| 15 min | 2.17±0.17,a | 2.14±0.17,b | 2.17±0.05,ab | 2.12±0.18,a |
| 1 hr | 2.28±0.17,b | 2.24±0.11,b | 2.51±0.18,b | 2.18±0.10,b |
| 2 hrs | 2.45±0.14,b | 3.03±0.10,c | 3.33±0.10,c | 2.48±0.02,bc |
| 6 hrs | 3.31±0.10,c | 3.56±0.20,d | 3.80±0.12,cd | 2.94±0.14,cd |
| 12hrs | 3.72±0.14,c | 3.80±0.16,d | 4.09±0.17,d | 3.13±0.04,d |
| 24hrs | 3.89±0.19,c | 4.71±0.08,e | 4.61±0.27,e | 4.27±0.02,e |

Begomovirus are a major threat to cultivation of crops in most tropical and subtropical regions of the world [26]. Begomovirus have either one (monopartite, DNA-A-like) or two (bipartite, DNA-A and DNA-B) circular ssDNA molecules of nearly 2800 nucleotides each. Each of these molecules is encapsulated by geminate particles of 22 nm* 38nm size assembled from the coat protein (CP) [10]. Some diseases namely, Yellow vein mosaic disease (YVMD), okra enation leaf curl disease (OELCD) and okra leaf curl disease (OLCD), caused by distinct begomovirus leads to menace in okra cultivation in India [12–14, 27].

*B.tabaci* ingests the Begomoviruses particles while feeding on the phloem sap of infected plants using stylets. The virions after passing the food canal ultimately reaches the midgut of its vector where the translocation of the begomovirus takes place through the filter chamber. Subsequently the virions are released into haemolymph from where they translocate into the salivary glands until further egestion. This spectacular interaction of begomovirus with *B. tabaci* has drawn worldwide attention since the past five decades [10].

Interestingly, transmission of begomovirus by whiteflies are reported to be species/strain specific i.e. transmission of a particular begomovirus occurs with different level of efficacy or specificity for each different species or strain [16]. For example, transmission of *tomato yellow leaf curl china virus* (TYLCCNV) in tomato and tobacco is transmitted by MEAM I and Asia II 3 cryptic species of *B.tabaci* but not Asia II I or MED [28, 29]. In this context, the present experiment deals with the comparative study of virus titre (YVMV and OELCuV) ingested and egested by *B.tabaci* Asia I and Asia II 5 at different time interval. Significant amount of variation was noted in virus uptake by both the genetic groups of whitefly. We did not detect any virus titre in whitefly samples immediately (1 min) after the virus exposure. Very low level of the virus titre at 1 minute of feeding which could not be detected by rt-PCR could be a possible reason. Also, the need of buffer time by *B.tabaci* to direct its stylet in the phloem tissue may justify no detection of virus copies immediately after feeding begins [30]. Previous studies using conventional PCR based technology suggests Asia I requires a minimum of 20 minutes of acquisition access period (AAP) to acquire virus (YVMV) [31] however, what we observed from our study is that *B.tabaci* Asia I could acquire YVMV just after 5 minutes of feeding.

The results also revealed that Asia I genetic group of *B.tabaci* acquired more titre of both the viruses in comparison to Asia II 5 in the initial phases of active feeding. Subsequently, after 24 hours of continuous feeding Asia II 5 acquired more of YVMV whereas, Asia I ingested more OELCuV. As, ability of virus transmission by whitefly species is positively associated with the virus titre in their body [16] it would not be wrong to state that Asia I could be a more efficient transmitter for OELCuV and Asia II 5 for YVMV on okra plants. Variation noted in virus uptake may be due to the variation of virus titre in the source plant itself [32]. To eliminate this variation in viral DNA in source plant we collected top leaves of same physiological age from both the plants. Viral copy number in the source plants of YVMV and OELCuV differed non-significantly. However, the exact molecular mechanism resulting in the differing ability of Asia I and Asia II 5 to acquire begomovirus still remains unclear and calls for future investigations.

Furthermore, egestion of virus copies by both the genetic groups were recorded at different time interval of feeding. Similar to virus acquisition, Asia II 5 inoculated higher titre of YVMV whereas Asia I inoculated higher titre of OELCuV at 24hr feeding exposure. Hence, it can be argued that the genetic group acquiring higher titre of virus egested higher amount during inoculation period. Another noteworthy fact is that copies of YVMV and OELCuV in detached leaves were less than the copies acquired by *B.tabaci* Asia I and Asia II 5 in 24 hrs. An explanation that could indeed rationalize this observation is that no further replication of both the begomovirus occurs in the detached leaves. Adding to this can be the inability of *B.tabaci* to inoculate exact amount of virus into the plant as the number of virions accumulating into the appropriate salivary components after circulation/replication might not be the same.

Acquisition and transmission of begomovirus by *B.tabaci* has been a major area of research throughout the world. However, most of the study has been conducted using conventional PCR technology, radioactive and non-radioactive probes. This present study utilizes absolute quantification method for determining the actual viral load acquired and inoculated by the two such genetic groups of whitefly, predominant in India. Moreover, results of the current investigation emphasises the comparative virus-vector relationship of two genetic groups of whitefly which will provide assistance in monitoring, rapid screening of resistant varieties and further controlling the diseases caused by these virus.

## Supporting information

**S1 Fig. Ingestion and egestion process of YVMV and OELCuV by single *B.tabaci* (both Asia I and Asia II 5) using detached leaf assay.**
(DOCX)

## Acknowledgments

The first author thankfully acknowledges Bidhan Chandra Krishi Viswavidyalaya (ICAR accredited State Agricultural University) for providing the University Research Scholarship to carry out the research work.

## Author Contributions

**Conceptualization:** Mritunjoy Barman, Snigdha Samanta.

**Formal analysis:** Mritunjoy Barman, Snigdha Samanta, Swati Chakraborty.

**Investigation:** Arunava Samanta, Jayanta Tarafdar.

**Methodology:** Mritunjoy Barman, Swati Chakraborty.

**Resources:** Jayanta Tarafdar.

**Supervision:** Mritunjoy Barman, Arunava Samanta, Jayanta Tarafdar.

**Writing – original draft:** Mritunjoy Barman, Snigdha Samanta.

**Writing – review & editing:** Arunava Samanta, Jayanta Tarafdar.

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
