## [Decision Letter · Decision Letter 0]

4 Feb 2022

PONE-D-21-40408Copy number variation of two begomovirus acquired and inoculated by different cryptic species of whitefly, Bemisia tabaci in OkraPLOS ONE

Dear Dr. Tarafdar,

Thank you for submitting your manuscript to PLOS ONE. After careful consideration, we feel that it has merit but does not fully meet PLOS ONE’s publication criteria as it currently stands. Therefore, we invite you to submit a revised version of the manuscript that addresses the points raised during the review process.

We look forward to receiving your revised manuscript.

Kind regards,

Rajarshi Gaur

Academic Editor

PLOS ONE

Journal Requirements:

Reviewers' comments:

Reviewer's Responses to Questions

**Comments to the Author**

1. Is the manuscript technically sound, and do the data support the conclusions?

Reviewer #1: Partly

Reviewer #2: Yes

2. Has the statistical analysis been performed appropriately and rigorously? 

Reviewer #1: Yes

Reviewer #2: Yes

3. Have the authors made all data underlying the findings in their manuscript fully available?

Reviewer #1: Yes

Reviewer #2: Yes

4. Is the manuscript presented in an intelligible fashion and written in standard English?

Reviewer #1: Yes

Reviewer #2: No

5. Review Comments to the Author

Reviewer #1: This study reports the variations in begomoviral acquisition and transmission by Asia I and Asia II 5 species of whiteflies. Real time PCR method is used at periodic intervals to quantify viral loads during ingestion and egestion by whiteflies. Significant variations are observed in virus uptake and transmission by both species for yellow vein mosaic & okra leaf curl virus. These variations indicate Asia I is better transmitter for OELCuV and Asia II 5 for YUMV which can significantly affect the virus pandemics. Overall, the manuscript is clearly written, data is clearly represented and results are explained well. However, I have few major questions which should be addressed before consideration for submission.

1- The authors did not mention about the sample size for example the number of whiteflies used for ingestion assay as well as for egestion assay. Please include these details for sample size for each timepoint in methods section.

2- There is no information about the sex ratio of the number whiteflies used for ingestion and egestion assay. As there are reports stating females feed and inoculate more over male whiteflies therefore considered better transmitters over males. If male to female ratios is not kept consistent then it can significantly affect the observations. Including this relevant information will add more vigor to the present observations.

3- Did authors check the expression of any endogenous genes to normalize basal levels?

4- Please explain term inoculation period which is used throughout the manuscript.

5- The authors have focused on acquisition and egestion of the virus but did not discuss about retention by whiteflies. Was there any data or observation made about retention of the virus?

6- Since there is significant amount of variation observed in virus uptake. Did authors perform any quantification by real time PCR to normalize the initial amounts of viruses used for feeding?

7- The sequences for the accession numbers OL743532 & OL743533 are not found in NCBI GenBank. Please update.

8- In figure 4 D data point for 15 min has large deviation than others. Please explain.

9- Did authors tried any other method of virus detection like western blot to add strength to this real time data?

Reviewer #2: the article might be of interest to a broad audience who might want an introduction to the management on crop improvement. I have enjoyed reading the entire manuscript except at a few places with typo errors and some sentences are very long and not appropriate. I advise authors to read again and correct them wherever necessary.

6. PLOS authors have the option to publish the peer review history of their article (what does this mean?). If published, this will include your full peer review and any attached files.

Reviewer #1: No

Reviewer #2: **Yes: **Ritesh Mishra

---

## [Author Response · Author response to Decision Letter 0]

14 Feb 2022

Reviewer #1:

This study reports the variations in begomoviral acquisition and transmission by Asia I and Asia II 5 species of whiteflies. Real time PCR method is used at periodic intervals to quantify viral loads during ingestion and egestion by whiteflies. Significant variations are observed in virus uptake and transmission by both species for yellow vein mosaic & okra leaf curl virus. These variations indicate Asia I is better transmitter for OELCuV and Asia II 5 for YUMV which can significantly affect the virus pandemics. Overall, the manuscript is clearly written, data is clearly represented and results are explained well. However, I have few major questions which should be addressed before consideration for submission.

Comment 1: The authors did not mention about the sample size for example the number of whiteflies used for ingestion assay as well as for egestion assay. Please include these details for sample size for each timepoint in methods section.

Response: Thank you for your valuable comments. We have taken individual whitefly sample of the both the genetic groups for quantification of YVMV and OELCuV. The use of individual whitefly sample for this study has been mentioned in line no. 122, 126 and 132 under material method section. For better understanding we have included a pictorial schematic template for the entire process of ingestion and egestion by both the genetic groups of B.tabaci as supplementary file (S1 Fig).

Comment 2: There is no information about the sex ratio of the number whiteflies used for ingestion and egestion assay. As there are reports stating females feed and inoculate more over male whiteflies therefore considered better transmitters over males. If male to female ratios is not kept consistent then it can significantly affect the observations. Including this relevant information will add more vigor to the present observations.

Response: Thank you for your valuable comments. We have used a single whitefly adult female for studying the ingestion and egestion assay. This is mentioned is line no. 122. 

Comment 3: Did authors check the expression of any endogenous genes to normalize basal levels?

Response: Thank you for your valuable comments. However, I would like to add that by running RT-PCR for the virus we can determine the viral load. The earlier cycle threshold will indicate more virus copies were present in the initial sample and caused more severe infection. For example, if we have 3 patients samples and one healthy control. We get Ct values: 8, 10, 30 and 0 (no signal) for control. These results will clearly indicate that patient 1 (with CT value 8 has the most severe infection) as the sample was amplified to be detectable only after 8 cycle of amplification meaning very high initial copy number, patient 2 with CT value 15 has a mild infection and the last patient have very few copies of the virus that was amplified only after 30 cycles. If we further run PCR products on the gel we might even see the difference if they are explicit. i.e. the first patient band would be brighter (which indicates qualitative PCR). However, RT-PCR allows us to distinguish very subtle changes. Accordingly, we think it will not be very helpful to use any housekeeping gene when working with the virus material so we can operate directly with the CT values. There are few techniques where we can convert CT values to the viral copy numbers, this was determined in the literature for most known viruses usually by building the linear or log curve of the known copy numbers and projecting the CT values to the curve. However, it is to be mentioned that for relative gene expression / relative quantification we could use any endogenous control like actin, EF etc. 

Comment 4. Please explain term inoculation period which is used throughout the manuscript.

Response: Thank you for your valuable comments. We would like to mention that the time period wherein the virus is egested with the saliva into the plant phloem by whitefly is referred to as inoculation period in the manuscript.

Comment 5. The authors have focused on acquisition and egestion of the virus but did not discuss about retention by whiteflies. Was there any data or observation made about retention of the virus?

Response: Thank you so much for your valuable comments. YVMV and OELCuV are typical persistent, non-propagative begomovirus. There are certain reports of TYLCV replication in whitefly however, no evidence have been found regarding the replication of YVMV and OELCuV occurring in whitefly. Hence only the acquisition and egestion of the virus has been focused in this study. However, we would like to add that we are currently studying the same phenomenon for these begomoviruses but we did not find any evidence of replication occurring in whitefly. 

Comment 6. Since there is significant amount of variation observed in virus uptake. Did authors perform any quantification by real time PCR to normalize the initial amounts of viruses used for feeding?

Response: Thank you so much for your valuable comments. To overcome the variation in viral DNA in source plant, the top leaf of plants of same physiological age were used. The viral copies in the source plants of YVMV and OELCuV were also quantified in real time PCR which resulted almost equal values. This has been mentioned in line no. 129. 

Comment 7. The sequences for the accession numbers OL743532 & OL743533 are not found in NCBI GenBank. Please update.

Response: Thank you so much for your valuable comments. We have updated the accession numbers as suggested and now it is available in NCBI.

Comment 8. In figure 4 D data point for 15 min has large deviation than others. Please explain.

Response: We appreciate the reviewer’s critical observation of Fig 4D as the figure data (SEM) were wrongly pasted from excel. In Fig 4D the SEM-values of Ct means were mistakenly plotted which shows huge variability at the time frame. The graphs for Fig 4D has been replotted with correct values and incorporated in the revised manuscript.

Comment 9. Did authors tried any other method of virus detection like western blot to add strength to this real time data?

Response: We agree with the reviewer’s suggestion. This would be an interesting study. However, this was beyond the scope of the current study. We will try to address this suggestion in our future studies.

Reviewer #2: The article might be of interest to a broad audience who might want an introduction to the management on crop improvement. I have enjoyed reading the entire manuscript except at a few places with typo errors and some sentences are very long and not appropriate. I advise authors to read again and correct them wherever necessary. 

Response: Thank you so much for your valuable comments. We have re-read the entire manuscript and made the necessary changes accordingly. We have revised the manuscript and tried to improve it as much as possible.

---

## [Decision Letter · Decision Letter 1]

14 Mar 2022

Copy number variation of two begomovirus acquired and inoculated by different cryptic species of whitefly, Bemisia tabaci in Okra

PONE-D-21-40408R1

Dear Dr. Tarafdar,

We’re pleased to inform you that your manuscript has been judged scientifically suitable for publication and will be formally accepted for publication once it meets all outstanding technical requirements.

Kind regards,

Rajarshi Gaur

Academic Editor

PLOS ONE

**Comments to the Author**

1. If the authors have adequately addressed your comments raised in a previous round of review and you feel that this manuscript is now acceptable for publication, you may indicate that here to bypass the “Comments to the Author” section, enter your conflict of interest statement in the “Confidential to Editor” section, and submit your "Accept" recommendation.

Reviewer #1: All comments have been addressed

Reviewer #2: All comments have been addressed

2. Is the manuscript technically sound, and do the data support the conclusions?

Reviewer #1: Yes

Reviewer #2: Yes

3. Has the statistical analysis been performed appropriately and rigorously? 

Reviewer #1: Yes

Reviewer #2: N/A

4. Have the authors made all data underlying the findings in their manuscript fully available?

Reviewer #1: Yes

Reviewer #2: Yes

5. Is the manuscript presented in an intelligible fashion and written in standard English?

Reviewer #1: Yes

Reviewer #2: No

6. Review Comments to the Author

Reviewer #1: The revised manuscript version is significantly improved and addressed major concerns raised from the reviewers.

Reviewer #2: Dear Authors

Thanks for the corrections as suggested but again I will advise to read again carefully and correct them wherever necessary.

7. PLOS authors have the option to publish the peer review history of their article (what does this mean?). If published, this will include your full peer review and any attached files.

Reviewer #1: No

Reviewer #2: No

---

## [Editor Report · Acceptance letter]

17 Mar 2022

PONE-D-21-40408R1 

Copy number variation of two begomovirus acquired and inoculated by different cryptic species of whitefly, *Bemisia tabaci* in Okra 

Dear Dr. Tarafdar:

I'm pleased to inform you that your manuscript has been deemed suitable for publication in PLOS ONE. Congratulations! Your manuscript is now with our production department. 

Kind regards, 

on behalf of

Professor Rajarshi Gaur 

Academic Editor

PLOS ONE